# Improving food security of farming households in Nigeria: Does broiler outgrowers' program make any difference?

**Olabisi Damilola Omodara**⊙*ⓞ, **Akinsola Temitope Oyebanji**ⓞ, **Oluwemimo Oluwasola**

Department of Agricultural Economics, Obafemi Awolowo University, Ife, Nigeria

ⓞ These authors contributed equally to this work.
* omodarao@oauife.edu.ng, omodarao@outlook.com

## Abstract

The purpose of this research was to examine the impact of outgrowers' programs on the food security of smallholder poultry farming households in Osun State. Using multi-stage sampling technique, a structured questionnaire was designed to collect information from beneficiaries and non-beneficiaries of outgrowers' programs in the study area. Descriptive statistics, food security index, and Heckman's selection model were used to analyze the data. The results revealed that outgrowers and non-outgrowers were on the average, 49 and 45 years old, with about 12 and 8 years of experience in poultry farming, respectively. The Poultry Farmers' Association was represented by 97% of outgrowers and 47% of non-outgrowers. There were four major broiler outgrowers' programs existing in the area. The Anchor-Borrower Outgrowers' program and Osun Broiler Outgrowers' program adopted a fixed contract model, whereas, the Dayntee Farm and GS Farm outgrowers' programs employed a semi-fixed contract model. The incidence of food insecurity was 18% for out-growers and 35% for non-outgrowers, with food insecurity depth and severity being 0.025 and 0.033 for outgrowers and 0.134 and 0.52 for non-outgrowers, respectively. The study found a significant difference in outgrowers' perceptions of food insecurity as well as their coping strategies. The major perceived indicators of food insecurity were inadequate resource endowment (MD = 0.758, $p<0.01$) and consumption of low-cost food (MD = 0.0658, $p<0.01$). Food acquisition on credit (WMS = 1.700), meals adjustment (WMS = 1.425), and cooking methods' modification (WMS = 1.875) strategies were adopted to cope with food insecurity. Participation in the outgrowers' program was influenced by membership of Poultry Farmers' Association, credit access and flock sizes and the significant predictors of food security among the poultry farming households were outgrowers' participation, household size, gender, marital status and credit access. It was therefore inferred that out-growers were considerably more food secure than the non-outgrowers, encouraging the need to scale up the program in the poultry industry. Introduction of flexible regulations and reproductive education would make the program more rewarding to the poultry farming households.

**Data Availability Statement:** All relevant data are found in this link. https://data.mendeley.com/datasets/ffffhxnjmc/1. DOI:10.17632/ffffhxnjmc.1

**Funding:** The author(s) received no specific funding for this work.

**Competing interests:** The authors have declared that no competing interests exist.

## Introduction

It is well known that humans need food in order to survive. When all people, at all times, lack physical and financial access to sufficient, secure, and nourishing food that meets their dietary needs and food preferences for an active and healthy life, food insecurity results [1]. According to [2], around 795 million people globally suffer from chronic undernourishment, and Africa is responsible for more than 60% of the world's food insecurity. With a hunger index score of 28.3, Nigeria, a significant African nation in the Sub-Sahara region, solely accounts for 10% of African households who are food insecure [2, 3]. This score foretells a serious hunger and food insecurity level for the country compare to other African countries with similar economies [2]. [4] observed that rural households, where agriculture is the mainstay of the economy, account for more than 70% of Nigeria's food insecurity problem. Therefore, promoting agriculture is essential to addressing Nigeria's food insecurity challenge.

Over 80% of Nigeria's population is employed in agriculture and relies on it for livelihood [5]. Smallholder poultry farming households, in particular, dominate the poultry sub-sector, raising birds in backyards, cages, and housings with limited farm infrastructure and technology [6]. These households rear chickens with the assistance of other family members who live with them, share resources, and eat from the same pot, and under the same roof [7]. By producing meat and eggs, which are essential proteins, and earning income to support farm households, poultry farming households play significant roles in Nigeria's attainment of the sustainable development goal of eradicating hunger. However, the system is plagued by a number of problems, including poor financing and market price volatility, which restrict the ability of farming households to be food secure [8–11].

One of the major market-based financing initiatives in Nigeria is the outgrowers' program. This program is designed to improve farm revenue, socio-economic status, and food security of the Nigerian farmers who raise broilers [12–14]. In other words, broiler outgrowers' program is a contract farming system designed to address the enduring issues with broiler farm financing and marketing in Nigeria. The goal of the program is to supply broiler farmers with birds, cash, feeds, and market supports. It then creates a connection between smallholder poultry farmers (out-growers) and reputable large-scale processors (off-takers) in order to improve the capacity utilization of the coordinated industries.

According to the terms of the outgrowers' contract, the off-taker company is required to pay the working capital and operating expenses associated with raising broiler chickens for a predetermined period, typically six weeks, and purchase the finished birds at the predetermined weight, price, and quantity [12]. Outgrowers' program is intended for active broiler farmers who are members of the Poultry Farmers' Associations (PFAs), and have the ability to meet the minimum bird capacity specified for broiler production in the program guidelines [12]. The structure of the outgrowers' program enables individual participants, groups of participants, and State governments to serve as program anchors [15]. One of the States in Nigeria where the Private, State, and Federal outgrowers' programs coexist in the broiler sub-sector is the Osun State. Osun Broilers Outgrowers' program (OBOP) and Anchor-Borrowers Outgrowers' program (ABOP), respectively, have been in effect in the State since 2011 and 2018, respectively. There is a proof that over the course of the OBOP's first three years of operation, 162 poultry producers profited, investing a total of around ₦788 million and raising more than 5 million birds [16].

Several studies have assessed the effects of the outgrowers' program in Nigeria with mixed results. [9, 11] highlighted that as non-outgrowers primarily profited from price variations in the market, the outgrowers have a higher likelihood to be food insecure than the non-outgrowers. Additional evidence supports the idea that contract's conditions have a big impact on

how outgrowers' programs affect participants' welfare [17]. Contrarily, studies by [17–20] claimed that farmers who took part in outgrowers' programs earned more income and were more likely to be food secure than the non-participating farmers. There is a paucity of information on the influence of the outgrowers' program on the food security of the poultry farming households in Nigeria, despite the fact that many works of literature on the subject concentrate on the agricultural sector. To this end, this study investigated the socio-economic attributes of broiler outgrowers and non-outgrowers, as well as the forms and contract models used in the State's outgrowers' programs. It further evaluated the food security of the broiler outgrowers and non-outgrowers; describes how each group perceives and copes with food insecurity; and determined the impact of participation in the poultry outgrowers' program on the food security of participating smallholder poultry farmers in the study area.

## Materials and methods

This study involved a cross-sectional survey of poultry farmers between April and July 2021 in Osun State. The State is located in the South-western Nigeria with a geographic area of around 9,251km and a population of 3,416,959 people. It is situated between latitudes 5˚58'N and 8˚ 07'N of the equator and longitude 04˚00 E and 05˚05'E of the Green meridian [21]. Osun State has 30 Local Governments Area with three agricultural zones which are Iwo, Osogbo and Ife/ Ijesha. It is bounded in the North by Kwara State, in the East partly by Ondo and Ekiti States, in the South by Ogun State and in the West by Oyo State. The State has a tropical humid climate with consistent temperatures throughout the year. The dry season occurs between November and March, while the rainy season occurs between April and October, with a mean annual rainfall of roughly 1,000 mm and a mean annual temperature ranging from 21.1˚C to 31.1˚C, which is favorable for poultry farming.

According to a report from the Osun State Ministry of Agriculture and the Central Bank of Nigeria, as of the first quarter of 2021, Osun State has about 42 and 105 ABOP and OBOP recipients, respectively. Some of these farmers took part in up to two outgrowers' programs at a time, the majority of which were OBOP and ABOP. The sample size was determined following the 95% confidence interval formula proposed by [22]. The outcome was 107 samples for outgrowers and non-outgrowers each. However, only 60 outgrowers and 60 non-outgrowers were sampled due to attrition and the inaccessibility of outgrowers.

Using a structured questionnaire, a multi-stage sampling technique was adopted to gather primary data for this study. At the first stage, we purposefully chose broiler farmers only because the poultry-based outgrowers' initiative solely targeted broiler farmers in Nigeria. The list of outgrowers' program beneficiaries from the Central Bank of Nigeria and the Osun State Ministry of Agriculture were then obtained for the study. At the second stage, we purposely selected six (6) Local Government Areas (LGAs) from 30 LGAs in the State due to the concentration of broiler growers in the area. At the third stage, we randomly selected 10 outgrowers and 10 non-outgrowers from each of the six selected Local Government Areas, to make an aggregate of 60 outgrowers and 60 non-outgrowers, and a total 120 respondents were sampled. However, only 40 outgrowers and 60 non-outgrowers of the total sampled respondents provided enough data for statistical analysis.

Poultry farming information were gathered from the poultry household heads based on their farm production activities during the first quarter of 2021. Socio-economic characteristics of farmers; types of outgrowers' programs; bird capacity and flock size; broiler production cost and income; and perception and coping mechanisms for food insecurity were collected. Using a Stata package version 15, the data were analyzed by descriptive statistics, the Foster-Greer-Thorbecke (FGT) food security index, and the Heckman's selection model. For face

validity of the instrument used, the questions were framed in line with previous similar studies and were given to experts to view vis-a-vis the research objectives. The content validity was carried out by adapting the questionnaire to the United Nation's Food Frequency question-naires (FFQ), in line with the adequacy of international literature on food security while the internal validity of the instrument was done by sampling 10 outgrowers and non-outgrowers each from non-sampled area of Ile-Ife and tested for internal consistency. The result showed a Cronbach alpha coefficient higher than the required minimum ($>0.60$). The socio-economic characteristics of the respondents, types of the outgrowers' program, the perception of food security among broiler farmers, and their coping mechanisms were then identified using descriptive statistics, such as frequency count, percentage, arithmetic mean, and weighted mean score. A Weighted Mean Score (WMS) mechanism was also deployed to obtain the average scores for the food security perception and coping strategies of broiler farming households while Food Security Index and Heckman's selection model were employed to determine the food security of outgrowers and non-outgrowers.

## Food security index model specification

Using the Food Security Index model, we determined the food security index of outgrowers and non-outgrowers by estimating the per capita household food expenditure. Following [20, 23], we derived the food security index by dividing the per capita $n^{th}$ household food expenditure by two-thirds mean per capita of all the households' expenditure as given in Eqn (1) in S1 File where Z = Food security index/line. If $Z \geq 1$, the household is food secure; If $Z < 1$, the household is food insecure. We then categorized households whose per capita monthly food expenditure were at least two-third of the mean per capita monthly food expenditure as food secure. However, we regarded those households whose per capita monthly food expenditure were less than two-third of the mean monthly per capita food expenditure as the food insecure households. We further employed the Foster, Greer and Thorbecke index used by [20, 23] to determine the food security index, the depth, gap and severity of food insecurity. The model is given in the Eqn. (2) in S1 File where: F is the food security measure (incidence, depth, severity); α symbolizes poverty aversion parameters, it takes values of 0, 1 or 2; n is the total number of households in the population; q is the number of households below the food security line (head count); $y_i$ symbolizes per capita food expenditure in increasing order for all households; z is the food security line ($\frac{2}{3}$ mean per capita of all households expenditure).

## Heckman's selection model

We employed Heckman's selection model to analyze the impact of outgrowers' program participation on household food security. We considered this model suitable due mainly to its capacity to correct for the non-random bias associated with the selection of poultry farmers for the outgrowers' programs. According [23, 24], the Heckman's selection model is a two-stage procedure. We employed Probit model at the first stage to determine the factors influencing participation of poultry farming households in the outgrowers program. The model is given as Eqn. (3) in S1 File where $d_i$ is the participation status of household; di = 1, for the outgrowers and 0 for the non-outgrowers. Φ(.) is the cumulative standard normal distribution; $Z_i$ is the observable socio-economic variables of the farming households including PFA membership, flock size, household head experience, age and gender of the household head; and $\gamma'$ is the the coefficient of predictor variable $Z_i$. The model has a likelihood function Eqn. (4) in S1 File where di is then estimated and inserted into the main welfare equation capturing food security.

As required by Heckman's selection model, at the second stage, we employed ordinary least square model to determine the effect of the outgrowers' program participation on the food security of the outgrowers. The model is given by Eqn. (5) in S1 File, where: $Y_i$ = food security index (two-third monthly per capita household expenditure); $X_1$ = household size (count); $X_2$ = marital status (1 = married, 0 = otherwise); $X_3$ = PFA membership; $X_4$ = household head poultry experience (years); $X_5$ = flock size (count); $X_6$ = bird selling price (naira); $X_7$ = household head access to credit facilities (1 = yes, 0 = otherwise); $X_8$ = Bird type (1 = broilers only, 0 = combined); $X_9$ = gender of the household head (1 = male, 0 = female); $U_i$ = random error; $\beta_0$ = constant term.

To account for the selectivity bias, we estimated Inverse Mills Ratio [24] which corrected for the correlation between the error term and the dummy variable by incorporating the expected value of selection error into the equation of the welfare outcome (food security). This is obtained by Eqn. (6) in S1 File. We then computed and fitted the Inverse Mill Ratio ($\omega_i$) in the OLS regression model from which the welfare equation becomes Eqns. (7) and (8) in S1 File where $d_i$ = 1 if the household participated in outgrowers' program and $d_i$ = 0 if otherwise.

## Ethical clearance for the study

The ethical clearance for this study was obtained from Health Research Ethic Committee (HREC), Institute of Public Health, Obafemi Awolowo University Ife. Each respondent signed the consent form after outlining the study's goals, potential benefits, confidentiality of the information provided as well as the right to revoke participation at any time.

## Results

### Socio-economic attributes of the broiler's outgrowers and non-outgrowers

For the outgrowers and non-outgrowers, the average age of the household heads was 49.38 ±9.15 years and 45.60±9.39 years, respectively (Table 1). Male farmers predominated the poultry industry because 78% of the outgrowers' households and 72% of non-outgrowers' households were headed by men. Averagely, 1.97±0.42 of the outgrowers and 2.03±0.55 of non-outgrowers were married with a mean household size of 6.48±1.74 and 5.52±2.40, respectively. Data in Table 1 further revealed that both the outgrowers and non-outgrowers, respectively, spent on the average, 11.75±6.32 and 8.23±5.36 years in schooling. The outgrowers had 11.7 ±6.32 years of experience, whereas, non-outgrowers had 8.23±5.34 years, however, there was

Table 1. Socio-economic characteristics of the smallholder poultry households.

| Variables | outgrowers | | Non-outgrowers | |
|---|---|---|---|---|
| | Mean | Std. Dev | Mean | Std. Dev |
| Age of HH head | 49.38 | 9.15 | 45.60 | 9.39 |
| Male HH head | 0.78 | 0.42 | 0.72 | 0.45 |
| Married HH head | 1.97 | 0.42 | 2.03 | 0.55 |
| Household size | 6.48 | 1.74 | 5.52 | 2.40 |
| Level of education of HH head | 6.87 | 1.41 | 5.90 | 2.22 |
| Year of experience of HH head | 11.75 | 6.32 | 8.23 | 5.34 |
| Poultry Farmers' Association | 0.98 | 0.16 | 0.47 | 0.50 |
| Access to credit | 0.83 | 0.38 | 0.52 | 0.51 |
| Credit source | 5.12 | 3.09 | 4.48 | 1.48 |

Source: Author's field survey, 2021.

no significant difference in poultry experience between both groups (t = 0.21, $p > 0.1$). In terms of credit access, 83% of the outgrowers and 52% of non-outgrowers had access to credit, and the majority of both groups acquired credit mainly from co-operatives groups. In all, about 98% of the outgrowers and 47% of non-outgrowers belonged to the Poultry Farmers' Associations (PFAs).

## Forms and bird releasing capacity of outgrowers programs

In Osun State, there were four main types of outgrowers' programs: the Anchor Borrowers' Outgrowers' Program (ABOP), Osun State Broiler Outgrowers' Program (OBOP), Dayntee Farm Outgrowers' Program, and the GS Farm Outgrowers' Program (Fig 1). Approximately, 48% of the outgrowers took part in the OBOP, 30% in the ABOP, 15% in the GS agricultural outgrowers' program, and 8% in the Dayntee farm outgrowers' program. ABOP and OBOP released a fixed number of broiler birds to the beneficiaries (Table 2). ABOP distributed 1,000 day old chicks, while OBOP distributed 2,000 day old chicks. GS Farm and Dayntee Farm, on the other hand, released varying numbers of broiler birds. GS Farms released 1,000 to 5,000 day old chicks, while Dayntee released a minimum of 2,000 day old chicks, the number may go as high as 5,000 day old chicks depending on the outgrowers' carrying capacity.

It was also noted that the Federal and Osun State governments drive cum fund ABOP and OBOP outgrowers' programs, respectively. Thus, both programs maintained a strict and rigid financing policy model. On the other side, Dayntee and GS Farm were funded by private firms and maintained a flexible outgrowers' contract policy. GS Farm outgrowers' initiative also uptook birds weightier than 2.0kg from non-outgrowers.

## Food security of the outgrowers and non-outgrowers' households

The food security indices for outgrowers and non-outgrowers in the research area were displayed in Table 3. The outgrowers' food security line was ₦11,098.59 ($10.22) while non-outgrowers' food security line was ₦10,188.97 ($10.31). According to the information in Table 3, more than three-quarters of the outgrowers (82.5%) and about two-third (65.0%) of non-

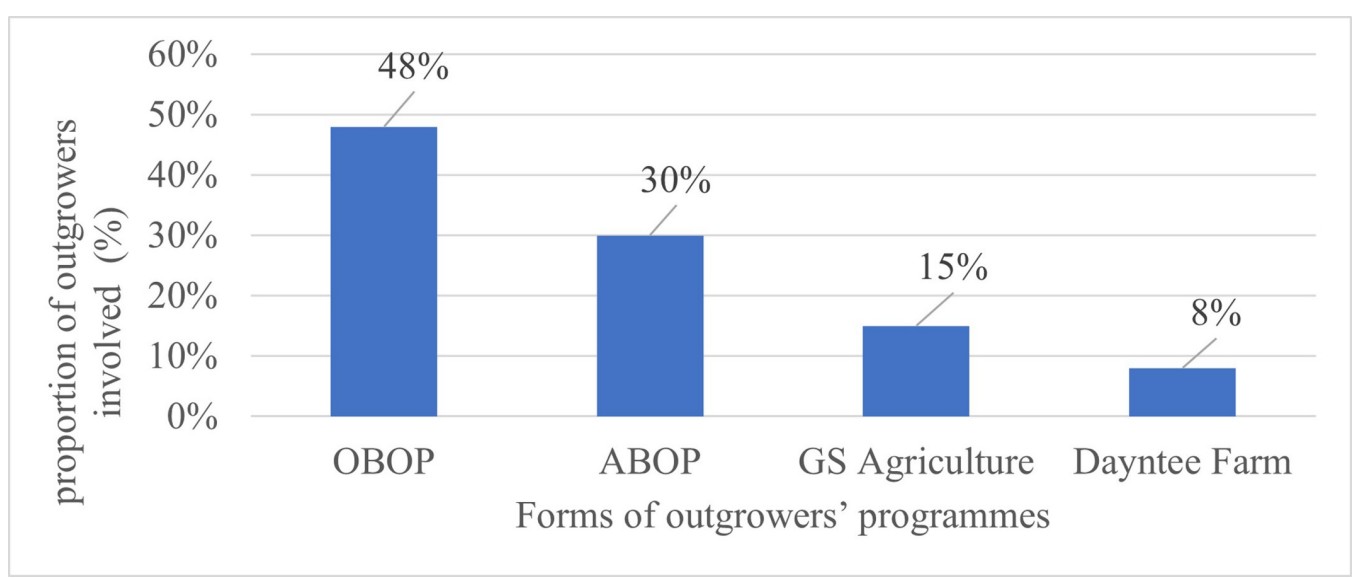

**Fig 1. Distribution of respondents by the existing forms of poultry outgrowers' programs and involvement rate in Osun State.**

**Table 2. Types of outgrowers in broiler production in Osun State.**

| Broilers' outgrowers type | Bird Capacity (count) | Contract Model | Sponsors |
|---|---|---|---|
| ABOP | 1,000 | fixed | Federal Government |
| OBOP | 2,000 | fixed | State Government |
| GS Farm limited | 1,000–5,000 | semi-flexible | private |
| Dayntee Farm | 2,000–5,000+ | semi-flexible | private |

Source: Field survey, 2021.

outgrowers had access to enough food but the outgrowers' food security situation differed significantly (t-statistics = 2.013, $p<0.05$) from that of the non-outgrowers.

We also learned from Table 4 that the outgrowers had 0.175 food insecurity incidence score whereas non-outgrowers had 0.350 incidence score, meaning that 17.5% of outgrowers and 35.0% of non-outgrowers had food expenditures per head that were below the threshold for food security. The outgrowers' depth of food insecurity was 0.025 in comparison to the non-outgrowers' depth of 0.134. This suggests that, in order for the food-insecure households among outgrowers and non-outgrowers to avoid food insecurity, they will need to spend an additional 2.5% and 13.4% of their family income on food. Similarly, the severity score of food insecurity for the outgrowers was 0.033, compared to the non-outgrowers' severity of food insecurity (0.052).

## Food insecurity perceptions among poultry household

In the past 4 weeks, both the outgrowers (WMS = 1.250±0.494) and non-outgrowers (WMS = 1.412±0.530) in Table 5 generally believed that they occasionally did not have enough food to eat (mean difference = 0.167, t = 1.040). In addition, we discovered that occasionally, both households belonging to the outgrowers (WMS = 1.025±0.158) and non-outgrowers (WMS = 1.133±0.343) skipped a meal because they did not have enough money to buy food (mean difference = 0.108, t = 0.397).

It was also realized that neither the outgrowers (WMS = 1.125±0.335) nor the non-outgrowers (WMS = 1.183±0.390) occasionally ate smaller meals than their households believed they required (mean difference = 0.058, t = 0.143). We also learned that in the past four weeks, both the outgrowers (WMS = 1.00±0.000) and non-outgrowers (WMS = 1.017±0.129) did not go a whole day and night without eating (mean difference = 0.167, t = 1.000). Table 5 further showed that while the outgrowers (WMS = 1.375±0.638) asserted that occasionally, their households were unable to eat certain types of food due to a lack of resources, non-outgrowers (WMS = 2.133±0.812) believed this assertion to be very often. The submission was backed by the outgrowers' mean difference (0.758) above non-outgrowers' mean, which was significant (t = -4.035). The outgrowers (WMS = 1.375±0.638) also agreed that, throughout the course of

**Table 3. Distribution of food security status of the outgrowers and non-outgrowers in the poultry outgrowers' program.**

| Categories | Food Secure | | Food Insecure | | Per capita household food expenditure (₦) | Total |
|---|---|---|---|---|---|---|
| | **Frequency** | **%** | **Frequency** | **%** | | |
| Outgrowers | 33 | 82.5 | 7 | 17.5 | 11,098.59 | 40(100) |
| Non-outgrowers | 39 | 65.0 | 21 | 35.0 | 10,188.97 | 60(100) |
| t-statistics | 2.013 | | 1.06 | | | |

Source: Field survey, 2021. Conversion rate [25]: $1.00 = ₦987.73

**Table 4. Food insecurity indices for the outgrowers and non-outgrowers.**

| Categories | Food Insecurity Indices | | |
|---|---|---|---|
| | Incidence ($F_0$) | Depth ($F_1$) | Severity ($F_2$) |
| outgrowers | 0.175 | 0.025 | 0.033 |
| Non-outgrowers | 0.35 | 0.134 | 0.052 |
| t-test | 1.06 | 0.39 | 1/09' |

Source: Field survey, 2021.

the previous four weeks, their households occasionally relied on any kind of inexpensive food to feed themselves. On the other hand, the non-outgrowers believed that very often, their households relied on any type of cheap food as households' food (WMS = 1.833±0.844). By and large, the non-outgrowers appear to have ongoing food insecurity crises based on the significant mean difference (0.658, t = -3.761) between them and the outgrowers.

## Coping mechanisms against food insecurity among poultry farming households

Modifying cooking techniques and using credit to buy food were the main coping mechanisms used by the participating households to combat food insecurity. Summarily, not less than 50% of the outgrowers used these methods (Table 6). Overall, 45% of the outgrowers changed their cooking techniques occasionally, 10% regularly, and 7% very often. Moreover, roughly 50% and 47% of the outgrowers borrowed food from friends and families (WMS = 1.700±0.853), and purchased food on credit (WMS = 1.700±0.91). Among these outgrowers, 27% occasionally purchased food on credit, while 17% did so on a regular basis. In a similar way, 32% occasionally borrowed money from friends and families, and 7% did so very often and regularly. A small percentage of outgrowers, or 32%, cut back on daily meals (WMS = 1.4250.712) and about 25% of the outgrowers did this occasionally. However, sales of household's assets were not widespread (20%) among the outgrowers' households (WMS = 1.325±0.724).

**Table 5. Households' perception of food insecurity.**

| Four weeks food security perception | Outgrowers | | Non-outgrowers | | Mean difference | t-statistics |
|---|---|---|---|---|---|---|
| | Weighted mean | Remark | Weighted mean (SD) | Remark | | |
| Have enough to eat | 1.250 (0.494) | occasional food insecure | 1.412 (0.530) | occasional food insecure | 0.167 | 1.04 |
| not able to eat kinds of food due to lack of resources | 1.375 (0.638) | occasional food insecure | 2.133 (0.812) | chronic food insecurity | 0.758 | -4.035* |
| skip a meal due to insufficient money to buy food | 1.025 (0.158) | occasional food insecurity | 1.133 (0.343) | occasional food insecurity | 0.108 | -0.397 |
| eat smaller meal than felt needed | 1.125 (0.335) | occasional food insecurity | 1.183 (0.390) | occasional food insecurity | 0.058 | 0.143 |
| rely on any kind of low cost food to feed | 1.375 (0.638) | occasional food insecurity | 2.033 (0.844) | Chronic food insecurity | 0.658 | -3.761* |
| Went a whole day and night without eating | 1.000 (0.000) | break even food security | 1.017 (0.129) | occasional food insecurity | 0.0167 | -1.00 |

Source: Field Survey, 2021. *, **, *** = significant at $p<0.01$, $p<0.05$, $p<0.1$ respectively

**Table 6. Coping mechanisms used against food security by poultry farmers.**

| Coping Mechanisms | Outgrowers (n = 40) | | | | | Non-outgrowers (n = 60) | | | | |
|---|---|---|---|---|---|---|---|---|---|---|
| | VO(%) | R(%) | O(%) | N (%) | WMS | VO(%) | R(%) | O(%) | N(%) | WMS |
| Obtained food on credit | 2.50 | 17.50 | 27.50 | 52.50 | 1.70 (0.853) | 6.67 | 16.67 | 45.00 | 31.67 | 1.36 (0.637) |
| Cut back on daily meals | 2.50 | 5.00 | 25.00 | 67.50 | 1.42 (0.712) | 13.33 | 35.00 | 33.33 | 18.33 | 1.36 (0.489) |
| Modified cooking methods | 7.50 | 10.00 | 45.00 | 37.50 | 1.87 (0.883) | 25.00 | 28.33 | 26.67 | 20.00 | 1.48 (0.653) |
| Sold asset | 5.00 | 2.50 | 12.50 | 80.00 | 1.32 (0.764) | 1.67 | 6.66 | 25.00 | 66.66 | 1.00 (0.00) |
| borrow from friends & relatives | 7.50 | 7.50 | 32.50 | 52.50 | 1.70 (0.911) | 5.00 | 15.00 | 43.33 | 33.33 | 1.32 (0.55) |

Source: Field Survey, 2021, VO = very often, R = regularly, O = occasionally, N = Never..

The non-outgrowers experienced the opposite. Table 6 shows that about 68% of households (WMS = 1.360±0.637) bought food on credit, with 2.5% involved the act of buying food on credit very often, 17% did so regularly, while 25% bought food on credit only occasionally. About 82% of households (WMS = 1.360±0.489) cut back on the amount of meals they ate each day. About 13% of those that reduced daily calorie intake did so very often, 35% regularly, and 33% only occasionally. Similarly, 80% of the non-outgrowers changed the way they cooked (WMS = 1.480±0.653). Overall, 28% modified cooking habits regularly, 25% very often, and 27% only occasionally. However, just 25% of the 33% of households sold assets on a regular basis, indicating that selling household assets (WMS = 1.000±0.000) was not a frequent habit among non-outgrowers. Also, borrowing from friends and families (WMS = 1.320±0.557) was another important strategy non-outgrowers adopted to cope with food insecurity as about 43% regularly, 33% borrowed only occasionally and 15% borrowed very often.

## Determinants of households' participation in outgrowers' programs

As shown in the result, PFA membership, credit access and flock sizes significantly (β = 1.923, $p<0.000$; β = 0.604 $p<0.1$; β = 0.001 $p<0.01$) increased the likelihood of participating in the outgrowers' programs (Table 7). The result shows that if there is a 1% increase in PFA membership, the likelihood that a broiler farmer will participate in the outgrowers' programs will increase by 192.3%. Similarly, flock size was positively correlated to outgrowers' participation. If there is a 1% increase in the flock size of broilers' farms, participation in outgrowers' programs will likely increase by 0.1%. Again, there is a strong correlation between having access to credit and taking part in the outgrowers' program. There is 60.4% livelihood that broiler farmer will participates in the schemes if credit access grows by 1%.

## Effect of outgrowers' programs on households' food security

The results in Table 7 demonstrate that household size significantly (β = -0.245, $p<0.01$) decreased food security while outgrowers' participation, gender of household head, marital status, and credit access significantly (β = 0.091, $p<0.01$; β = 0.248, $p<0.05$; β = 0.152, $p<0.1$ and β = 0.214, $p<0.1$) increased food security. The gender of the household head is positively correlated with per capita food expenditure, as seen in Table 7. Household per capita food expenditure would increase by 24.8%, if the gender of the household head changed by 1% (for instance, from female to male). Similarly, loan availability has a favorable impact on household food security. The amount of money households spend on food per person will grow by 21.4% for every 1% increase in credit accessibility.

According to Table 7, Contrarily, there was a strong positive correlation between food security and marital status. A 1% increase in marital status will cause a 10.2% rise in the

**Table 7. Heckman's selection model showing determinants of outgrowers' participation and its effect on household food security.**

| Variables | Selection/Determinants | | Outcome/Impact | |
|---|---|---|---|---|
| | Marginal effect | p-value | Coef. | p-value |
| Household size | -0.072 | 0.466 | -0.245 | 0.000*** |
| Marital status | -0.437 | 0.156 | 0.152 | 0.102* |
| PFA membership | 1.923 | 0.00*** | 0.016 | 0.958 |
| Years of experience | -0.029 | 0.421 | -0.005 | 0.639 |
| Flock size | 0.001 | .001*** | 0.0008 | 0.743 |
| Flock selling price | -0.001 | 0.449 | 0.0007 | 0.441 |
| Credit access | 0.604 | .081* | 0.214 | 0.075* |
| Age of the household head | 0.476 | 0.659 | | |
| Level of education | 0.069 | 0.489 | | |
| Bird type | | | -0.038 | 0.647 |
| Gender | | | 0.248 | 0.039** |
| IMR (participation) | | | -0.91 | 0.027*** |
| Constant | -2.286 | 0.468 | 3.282 | 0.003*** |
| athrho | -.78 (0.912) | 0.392 | | |
| lnsigma | -1.2 (0.311) | 0.000*** | | |
| Mean dependent var | 0.400 | SD dependent var | | 0.92 |
| Number of observations | 100 | Chi-square | | |
| Prob > chi2 | 0.000 | Akaike Crit. (AIC) | | 134.691 |

*** indicates a significant level at 1%

**indicates a significant level at 5%

* indicates a significant level at 10%

Source: Authors' computation, 2021.

participating households' per capita food expenditure. On the other side, there was an inverse relationship between household size and food security. According to the Heckman's model, the outgrowers' per capita food consumption will reduce by 24.5% for every 1% rise in household size. There was also a significant negative selectivity bias among households, as indicated by the negative correlation between participation (IMR) and food security. As a result, the mean per capital expenditure of outgrowers will rise by 91% due mainly to participating in the outgrowers' programs.

## Discussions

The prevalence of government-sponsored outgrowers' programs (ABOP and OBOP) among poultry farming households may be attributed to the intensive programs' public awareness-raising and sensitization campaigns via social networking sites, as well as the Poultry Farmers' Association's involvement in the eligibility requirements. The findings support [16] assertion that radio programs are crucial to Osun State's agricultural development programs.

The fixed contract model operating in government-sponsored outgrowers' initiatives implies that the outgrowers have no right to choose which farm inputs they want to receive from the sponsors. Unlike the government-sponsored outgrower scheme, both the GS Farm and Dayntee Farm outgrowers' models enabled outgrowers to choose the number of birds they would receive, but neither permitted them to control the quantity of complementing inputs that were delivered as part of the support package. This made both programs' bird supply somewhat flexible. In addition, only the anchor group has the authority to determine how

much of the complementary inputs will be delivered to the outgrowers in an equivalent amount. As a result, it could be inferred that the Dayntee Farm and GS Farm outgrowers' programs operate under a non-capsule outgrower policy.

It must be noted that the private-driven outgrower programs appear more favorable to farmers than their public counterparts. Aside this, the GS Farm's outgrowers' program is expanding its reach and strengthening the company's position in the poultry industry by providing market for the non-outgrowers' birds. Invariably, smallholder poultry farmers are more likely to see this contract model desirable than the ABOP and OBOP models. Sooner or later, if care is not taken, a considerable numbers of the beneficiaries of government-driven outgrowers' programs may attempt to switch allegiance to the private-driven outgrowers' programs. Given this narrative, the future of Nigeria's government-driven outgrowers' programs does not appear promising.

This study further shows that participation in the outgrowers programs was strongly influenced by outgrowers' PFA membership, meaning that smallholder poultry farmers will need to be encouraged to join PFAs in order to improve their chances of taking part in the programs. [20] reported the significance of PFA membership in participation in targeted programs, and our finding validates their findings. Similarly, outgrowers' initiatives favor broiler producers with access to credit and big flock sizes. Thus, it is affirmed that outgrowers' programs offer assistance to farmers who own production facilities but lack the necessary operating funds to raise chickens to maturity. This indicates that the Nigerian government's effort to finance and commercialise smallholder poultry farming through outgrower market initiative is succeeding in its mission. This result backs up claim of [10] who noted the importance of credit access to farmers' willingness to participate in farm support programs.

It should be highlighted that in comparison to the non-outgrowers' households, outgrowers' households had higher levels of food security and lower incidence, depth, and severity of food insecurity. However, under outgrowers' initiatives, large farming households will find it difficult to be food secure. It was observed that there was household size's threshold that guaranteed food security of the outgrowers notwithstanding the favorable link between marital status (married) and food security. There is no doubting that a large household stresses the limited resources of the home. As a result, outgrowers with large households may need to motivate a number of the household members to undertake productive ventures to increase and maintain the household's food security. It is interesting to note that benefiting and attaining food security objective of the outgrowers' program in the poultry sector calls for management of the outgrowers' household size. This report therefore corroborates findings of [7, 10] that credit access and household size are key determinants of food security in Nigeria. The negative selectivity bias that indicates that the outgrowers had greater food security than non-outgrowers demonstrates once more how efficient the outgrowers' program is at enhancing food security in Nigeria's smallholding chicken production. This submission supports findings of [17–19] who found out that the outgrowers have a higher tendency to be food secure but contradicts reports of [9, 11] that outgrowers' program exploits and does not improve food security status of the beneficiaries because the non-outgrowers take advantage of fluctuations in market prices to better their position in the market. However, due to differences in the sociocultural, ecological, and institutional settings, our findings might not be applicable in other climes and areas in Nigeria. Therefore, future research in Nigeria should take a cursory look at agro-ecological regions and the ways by which poultry outgrowers' programs have affected food security in the other regions and cultures in Nigeria.

## Conclusion and recommendations

The study evaluated the impact of outgrowers' programs on the food security of outgrowers' farming households in Osun State, Nigeria. Compulsory membership of Poultry Farmers' Association, farmers access to credit and availability of farm infrastructure with a bird capacity that meets the specified program requirement were key to participation in outgrowers' programs. Food security of outgrowers was strongly influenced by participation in outgrowers' program, gender, marital status of the household head, household size, and credit access. However, there was a threshold household size that guaranteed food security of outgrowers. Outgrowers also had higher level of education but education attainment did not influence food security of both the outgrowers and non-outgrowers. Further findings revealed that, despite the effects of this intervention in the poultry sector, households headed by women outgrowers were more likely to experience food insecurity than those headed by men.

One major challenge we encountered in this study was the non-readiness of the non-outgrowers to release information about their households despite consenting to be sampled. This was however minimal among the outgrowers. In cases of occurrence, respondents were assured of the confidentiality of their data. On this note, it was recommended that

i. The public-driven outgrowers' programs should be revised to accommodate more smallholder farmers in light of the fact that the outgrowers initiative promotes food security among broiler farmers. The programs will benefit the recipient farming households more if flexible regulations are implemented.

ii. It might be important to investigate the feasibility of modifying and expanding the privately-driven outgrowers model as part of the policy push for inclusive smallholder participation in Nigeria's poultry outgrowers' programs.

iii. Since the size of the household played a crucial role in the food security model, the anchor and development partners must stress the significance of family planning and reproductive education among smallholder poultry farming households.

iv. Promoting smallholder poultry farmers' membership of the poultry farmers' organization is necessary as it was a key component of their ability to participate in outgrowers' programs, which had major positive effects on outgrowers' household welfare.

## Supporting information

**S1 Appendix.**
(DOCX)

**S1 File.**
(DOCX)

## Author Contributions

**Conceptualization:** Olabisi Damilola Omodara, Oluwemimo Oluwasola.

**Data curation:** Olabisi Damilola Omodara, Akinsola Temitope Oyebanji.

**Formal analysis:** Olabisi Damilola Omodara, Akinsola Temitope Oyebanji.

**Investigation:** Akinsola Temitope Oyebanji.

**Methodology:** Olabisi Damilola Omodara, Akinsola Temitope Oyebanji.

**Project administration:** Akinsola Temitope Oyebanji.

**Resources:** Olabisi Damilola Omodara, Akinsola Temitope Oyebanji.

**Software:** Olabisi Damilola Omodara.

**Supervision:** Olabisi Damilola Omodara, Oluwemimo Oluwasola.

**Validation:** Olabisi Damilola Omodara.

**Visualization:** Olabisi Damilola Omodara.

**Writing – original draft:** Olabisi Damilola Omodara, Akinsola Temitope Oyebanji.

**Writing – review & editing:** Olabisi Damilola Omodara.

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
