## [Decision Letter · Decision Letter 0]

16 Nov 2022

PONE-D-22-26873Improving food security of farming households in Nigeria. Does broiler outgrowers’ programme makes any difference?PLOS ONE

Dear Dr. Olabisi,

Thank you for submitting your manuscript to PLOS ONE. After careful consideration, we feel that it has merit but does not fully meet PLOS ONE’s publication criteria as it currently stands. Therefore, we invite you to submit a revised version of the manuscript that addresses the points raised during the review process.

We look forward to receiving your revised manuscript.

Kind regards,

Mabel Kamweli Aworh, DVM, MPH, PhD. FCVSN

Academic Editor

PLOS ONE

Journal Requirements:

2. You indicated that ethical approval was not necessary for your study. We understand that the framework for ethical oversight requirements for studies of this type may differ depending on the setting and we would appreciate some further clarification regarding your research. Could you please provide further details on why your study is exempt from the need for approval and confirmation from your institutional review board or research ethics committee (e.g., in the form of a letter or email correspondence) that ethics review was not necessary for this study? Please include a copy of the correspondence as an ""Other"" file.

4. We suggest you thoroughly copyedit your manuscript for language usage, spelling, and grammar. If you do not know anyone who can help you do this, you may wish to consider employing a professional scientific editing service. 

5. We note that Figure 1 in your submission contain map image which may be copyrighted. All PLOS content is published under the Creative Commons Attribution License (CC BY 4.0), which means that the manuscript, images, and Supporting Information files will be freely available online, and any third party is permitted to access, download, copy, distribute, and use these materials in any way, even commercially, with proper attribution. For these reasons, we cannot publish previously copyrighted maps or satellite images created using proprietary data, such as Google software (Google Maps, Street View, and Earth). For more information, see our copyright guidelines: http://journals.plos.org/plosone/s/licenses-and-copyright.

Additional Editor Comments:

1. The entire manuscript is too lengthy and requires remarkable reduction please. The manuscript needs to be revised for typographical errors, please. The manuscript also needs to be presented using the PLOS ONE's style requirements please. The Results section needs to be separated from the Discussion section, please. Sub-headings should be deleted from the Discussion section. Please provide interpretation of your study results and possible explanations for your findings while comparing your study with results of other published works.

2. In your Methods section, please provide additional location information, including a much clearer map drawn using geographical coordinates as opposed to copying and pasting a map from the internet (this is unacceptable please). 

3. Please kindly provide evidence of ethical approval that you obtained for this study from a competent and recognized institutional review board especially since human subjects were involved in this study. Kindly provide the name of the ethical committee/institutional review board as well as approval number in the revised manuscript, please.

4. Please highlight the limitations of this study in the concluding part of the Discussion section. Also summarize the recommendations provided in the Conclusion section, please.

5. All the formulars and equations provided in the body of the manuscript should be deleted from the main manuscript and submitted as Supporting Information. 

6. Please include copies of the survey questions or questionnaires used in the study, in both the original language and English, as Supporting Information, or include a citation if they have been published previously.

Reviewers' comments:

Reviewer's Responses to Questions

**Comments to the Author**

1. Is the manuscript technically sound, and do the data support the conclusions?

Reviewer #1: Yes

Reviewer #2: Partly

Reviewer #3: No

2. Has the statistical analysis been performed appropriately and rigorously? 

Reviewer #1: Yes

Reviewer #2: I Don't Know

Reviewer #3: No

3. Have the authors made all data underlying the findings in their manuscript fully available?

Reviewer #1: Yes

Reviewer #2: No

Reviewer #3: Yes

4. Is the manuscript presented in an intelligible fashion and written in standard English?

Reviewer #1: No

Reviewer #2: Yes

Reviewer #3: No

5. Review Comments to the Author

Reviewer #1: The paper evaluates the impact of out grower programmes on food security of out grower households in Nigeria. The authors used appropriate data collection techniques, primary data survey method analysed using descriptive statistics, food security index and Heckman's selection model. The manuscript aims to find out if broilers out grower's programmes makes any difference in improving food security of farming household in Nigeria and reaches logical conclusions based on available data. The study shows that there is a significant difference in the perception of food insecurity among out grower and non out grower households with the out-grower household being at an advantage. The manuscript is technically sound and shows that reasonable sample sizes were used. However, sample was collected from Osun state and not all states in Nigeria. The title is not representative of the scope of study. Though, the statistical analysis from available data has been performed appropriately, it is important to note that the limitations of this study were not highlighted.

The data underlying the findings have not been withheld. The manuscript is presented in intelligent English. However, there are some concerns:

Editorial concerns: some lines in the abstract have spacing issues.

Introduction section is doubled spaced while abstract section single spaced

Typographical errors

Results and Discussion tittle at the end of one page while the body on another page

Table 1: Socioeconomic characteristics of smaller poultry households has part of the table at the end of one page and another part at the beginning of the next page

full meaning of acronym DAYNTEE and GS FARM not stated.

Table 4 has title on one page and body on another page.

Table 5 is discussed under Food insecurity perceptions among poultry households but the table itself is presented under Coping mechanisms against food insecurity among poultry farming households. The tables should be properly arranged and presented.

Conclusion and recommendation stated on one page and the body on another.

References stated on one page and the actual references on another page

Not all intext citations were listed on the Reference page.

The study is good despite its limited scope of study because it reveals that out grower programmes increases food security amongst broiler farmers and that household size is critical to the food security model while suggesting that small broiler farmers should be encouraged to join the poultry association, out grower programmes should be structured to accommodate the small poultry farmers as well as the introduction of reproductive education to farming households.

Reviewer #2: Interesting paper, however:

- Did you used a pre-validated questionnaire for data collection or you made your own? If you made your own, have you tested it before using it for data collection?

- Since you collected data from human subject, can you provide more details about the ethical approbation?

- Which software (and Version) have you used for data analysis? how did test the validity of your results?

- Please review the conclusion and make sure that it support your findings

Thank you and good luck with your submission

Reviewer #3: The entirety of the paper was not written in a manuscript format that permits thorough evaluation and direction.

Abstract

The objectives and purpose of the study were not presented in a manner that facilitates comprehension.

- There was a lack of understanding and information on the "Outgrowers program" and the significance of its impact on food security.

- No information regarding the statistics program and performed analysis.

- Abbreviations cited in the abstract were not specified or written out for clarity earlier in the abstract.

- The entire abstract was uninformative and failed to provide a strong understanding of what the researchers intended to achieve, how it was carried out, and how the data were created, in addition to drawing conclusions.

- The entire document has several typographical and grammatical mistakes.

- There were too many keywords.

Introduction

- The entire introduction requires significant modification and reorganization. 

- The study's justification is unclear because it was indicated elsewhere in the report that other researchers have previously worked on the same program. 

- Additionally, it contains numerous typographical and grammatical errors

Material and Methods

- Due to the fact that human subjects were interviewed while collecting data, the materials and procedures have a significant need for ethical approval.

- The tools and techniques used for data analysis are also unclear.

Results and Discussion:

- Results were not presented logically for reader's for comprehension.

- The entire document needs to be completely redone and formatted as a manuscript.

- It should also be subjected for a review by native English speaker.

6. PLOS authors have the option to publish the peer review history of their article (what does this mean?). If published, this will include your full peer review and any attached files.

Reviewer #1: No

Reviewer #2: No

Reviewer #3: No

---

## [Author Response · Author response to Decision Letter 0]

8 Dec 2022

REVIEWER 1

We have adjusted and reconciled table arrangement. It must be noted that the journal format expects every table to follow the first paragraph after mentioning. This is the reason one or two table still have to appear at border line. In addition, it was necessary to reduce the manuscript length, as such, author have to optimize the use of space. 

DAYNTEE and GS FARM are broiler processing firms’ names. We have written them in sentence case. (Dayntee Farm and GS Farm)

Table arrangement and numbering have been addressed

The separation of title and content has been addressed. 

All in-text citations have been referenced.

REVIEWER 2

Diligent efforts have been made to effect every correction raised by the reviewers research instrument design and validation. The content validity was by adapting the questionnaire to the United Nation’s Food Frequency questionnaires (FFQ), while the internal validity of the instrument was done by testing for internal consistency (Cronbach alpha coefficient)

Stata package version 15, already specified in the manuscript.

The conclusion and recommendation have been rewritten to reflect the findings from the study. 

A sub-section of the methodology explains how ethical consent was obtained for the study

REVIEWER 3

More information has been provided on outgrowers and their significance to food security. Please, kindly check page 3, Paragraph 1 for details

The statistical approach and analysis were duly provided. Kindly check page 6 and 7 for details.

All abbreviation cited in abstract have been addressed.

Thank you. The entire abstract have been rewritten and all concerned addressed

Thank you for the observation. The entire manuscript has been thoroughly formatted and proofread by a native English speaker.

The keywords have been reduced to 7, however, this journal permit up to 50 key words.

We appreciate this. Note that the entire manuscript has been reworked.

We appreciate this concern. We have adjust the justification section and make it clearer for understanding.

We appreciate this. Note that the entire manuscript has been reworked.

Thank you for your observation. It must be noted that all concerns about ethical issues have been addressed under a sub-section called “ethical consent for the study”. In my institution, only health related studies requires ethical approval from the institution authority. In agricultural economics, field survey involving farming households requires getting the consent of community leaders and trade union which we sought in this study. The community consent was obtained via a visit to the poultry farmers’ association in the commodity of study. A further survey right was obtained from the central bank of Nigeria that released the list of beneficiaries. Meanwhile, a copy of the consent form for the respondent was uploaded as additional file.

Thank you for the observation. The analytical tools for the study were specified in clear terms. We reported that descriptive, Food security index, and Heckman's selection model will be used for the study. Note that the food security index was meant to estimate household food security status which stood as dependent variable for Heckman's model. Secondly, Heckman's model is a 2-in-1 model. It addresses both the determinant of participation (Probit) and outcome (OLS regression) but also take into consideration the selectivity bias in the sample

Thank you for your comment. The result is separated form the discussion as such, the report has become clear for layman understanding.

This has been done

We appreciate this comment. Note that the entire manuscript has been proofread by a native English speaker in Canada.

Name: Jenifer Erhunmwunse. A nurse Practitioner at Micheal Garron Hospital, Toronto East Health Network, ON. E-mail: jennifercourage10@gmail.com; rhnj002@humblermail.ca.

---

## [Decision Letter · Decision Letter 1]

18 Jan 2023

PONE-D-22-26873R1Improving food security of farming households in Nigeria. Does broiler outgrowers’ programme makes any difference?PLOS ONE

Dear Dr. OMODARA,

Thank you for submitting your manuscript to PLOS ONE. After careful consideration, we feel that it has merit but does not fully meet PLOS ONE’s publication criteria as it currently stands. Therefore, we invite you to submit a revised version of the manuscript that addresses the points raised during the review process.

We look forward to receiving your revised manuscript.

Kind regards,

Mabel Kamweli Aworh, DVM, MPH, PhD. FCVSN

Academic Editor

PLOS ONE

Additional Editor Comments:

1. It is important to note that this manuscript cannot be published without the provision of an official ethics approval received from a recognized Institutional Review Board (IRB). This is because human subjects were involved in this study. In your response, kindly provide the name of the IRB that approved this study and the approval number in the revised manuscript.

2. Please provide additional supplementary data or a link to a public repository where the data can be accessed.

Reviewers' comments:

Reviewer's Responses to Questions

**Comments to the Author**

1. If the authors have adequately addressed your comments raised in a previous round of review and you feel that this manuscript is now acceptable for publication, you may indicate that here to bypass the “Comments to the Author” section, enter your conflict of interest statement in the “Confidential to Editor” section, and submit your "Accept" recommendation.

Reviewer #1: All comments have been addressed

Reviewer #2: All comments have been addressed

Reviewer #3: (No Response)

2. Is the manuscript technically sound, and do the data support the conclusions?

Reviewer #1: Yes

Reviewer #2: Yes

Reviewer #3: Partly

3. Has the statistical analysis been performed appropriately and rigorously? 

Reviewer #1: Yes

Reviewer #2: Yes

Reviewer #3: Yes

4. Have the authors made all data underlying the findings in their manuscript fully available?

Reviewer #1: Yes

Reviewer #2: Yes

Reviewer #3: No

5. Is the manuscript presented in an intelligible fashion and written in standard English?

Reviewer #1: No

Reviewer #2: Yes

Reviewer #3: Yes

6. Review Comments to the Author

Reviewer #1: Abstract: Despite the use of appropriate controls and sample sizes. The experiment does not seem to have appropriate replication. Sentence review needed.

Introduction: Study justification clear.

Materials and Methods: Appropriate

Ethical clearance: Appropriate

Results: The presentation of the results allows for ease of logical conprehension.

Conclusion and Recommendation: While the limitations of this study was explained. The author did not give any suggestions that can help in overcoming these limitations in future studies.

Typographical errors noted.

I wish you the best in your submission.

Reviewer #2: Thank you for the revisions made to the manuscript.

Please review the following questions before I can accept the revised version and recommend it for publication.

Is the the poultry farmers’ association in Osun State an approved Institutional Review Board? Do they have the mandate to give ethical approbation?

Again as your study involve human participants, Have you been able to receive an official IRB approval from an established and recognized IRB? It is important that you clarify this portion of your manuscript

Can you please provide the link for data access? I can't see the link to your study data, if the data are not in an repository can you share it as excel or any acceptable data management tools?

Regards

Reviewer #3: For simplicity of review and referencing, the authors should consider submitting their subsequent review in the appropriate manuscript format.

Ethical approval stated is not from a reputable institutional review board. Poultry farmer's association can grant permission but not ethical approval for a study. Given that this study involves human participants, could you submit documentation of the institutional review board's ethical permission or waiver you received? Please include the updated paper's approval or waiver number and the name of the institutional review board or ethics committee.

When citing studies in prose, could the authors consider stating the first author's name and year of publication and not and not just a reference number in the citation? An example was seen in the first paragraph of the introduction section, line 3, where the authors stated, " According to (2)."

Authors should make an effort to note the locations where the instrument's internal validity was tested. Was it in a similar environment to what was intended?

The authors are required to specify where the data that are stated to be available can be obtained.

7. PLOS authors have the option to publish the peer review history of their article (what does this mean?). If published, this will include your full peer review and any attached files.

Reviewer #1: No

Reviewer #2: **Yes: **Zoumana Isaac TRAORE

Reviewer #3: No

---

## [Author Response · Author response to Decision Letter 1]

28 Feb 2023

Reviewer’s comments

1.Despite the use of appropriate controls and sample sizes. The experiment does not seem to have appropriate replication. Sentence review needed.

Thank you. We have included multistage sampling technique used for sampling in the abstract. 

2.While the limitations of this study was explained. The author did not give any suggestions that can help in overcoming these limitations in future studies.

Thank you for your observation. We have made suggestions on how we overcame the limitations and how to overcome them in the future studies. 

3.Is the the poultry farmers’ association in Osun State an approved Institutional Review Board? Do they have the mandate to give ethical approbation

Actually, poultry association is not an ethical approbation authority, however, in as much as broiler anchor borrowers’ programme was practice by the members of the associations, it was necessary to obtain consent of the association in eliciting information about their operation. 

Again as your study involve human participants, Have you been able to receive an official IRB approval from an established and recognized IRB? It is important that you clarify this portion of your manuscript 

It must be noted that this study was carried out based on the consent obtained from the trade association and Central Bank of Nigeria however, due to the mandatory request for ethical clearance from the ethical issuing authority, we have submitted application for ethical clearance at the Institute of Public Health, College of Health Sciences, Obafemi Awolowo University, Ile-Ife, Nigeria. This process takes about 2 months for approval. 

A copy of the documents submitted is hereby attached to this document as evidence. 

We have decided to return our revised manuscript because the deadline given for the return of manuscript is 6th of March, 2022 meanwhile, the ongoing electoral process in the country had made the university system to be on break for 3 weeks, which means our application cannot be considered for approval until mid of May, 2023. Notwithstanding, if the evidence provided is not sufficient, we implore the publishing house to give us additional 2 months to process the ethical document for approval. 

4.Can you please provide the link for data access? I can't see the link to your study data, if the data are not in an repository can you share it as excel or any acceptable data management tools?  

https://data.mendeley.com/datasets/ffffhxnjmc/1. DOI:10.17632/ffffhxnjmc.1

5.When citing studies in prose, could the authors consider stating the first author's name and year of publication and not and not just a reference number in the citation? An example was seen in the first paragraph of the introduction section, line 3, where the authors stated, " According to (2)."

Thank you, all citation in pros have been changed to the numbering referencing style.

6.Authors should make an effort to note the locations where the instrument's internal validity was tested. Was it in a similar environment to what was intended?

Thank you for your observation. The location is Ile-Ife, a section of the study area. This has been indicated in the methodology.

---

## [Decision Letter · Decision Letter 2]

15 Mar 2023

PONE-D-22-26873R2Improving food security of farming households in Nigeria. Does broiler outgrowers’ programme makes any difference?PLOS ONE

Dear Dr. Omodara,

Thank you for submitting your manuscript to PLOS ONE. After careful consideration, we feel that it has merit but does not fully meet PLOS ONE’s publication criteria as it currently stands. Therefore, we invite you to submit a revised version of the manuscript that addresses the points raised during the review process.

We look forward to receiving your revised manuscript.

Kind regards,

Mabel Kamweli Aworh, DVM, MPH, PhD. FCVSN

Academic Editor

PLOS ONE

Additional Editor Comments:

It is important that the authors provide details of ethical approval from a reputable Institutional Review Board to protect the rights and welfare of human subjects involved in this study.  

Reviewers' comments:

Reviewer's Responses to Questions

**Comments to the Author**

1. If the authors have adequately addressed your comments raised in a previous round of review and you feel that this manuscript is now acceptable for publication, you may indicate that here to bypass the “Comments to the Author” section, enter your conflict of interest statement in the “Confidential to Editor” section, and submit your "Accept" recommendation.

Reviewer #1: (No Response)

Reviewer #3: (No Response)

2. Is the manuscript technically sound, and do the data support the conclusions?

Reviewer #1: Partly

Reviewer #3: Partly

3. Has the statistical analysis been performed appropriately and rigorously? 

Reviewer #1: Yes

Reviewer #3: Yes

4. Have the authors made all data underlying the findings in their manuscript fully available?

Reviewer #1: No

Reviewer #3: Yes

5. Is the manuscript presented in an intelligible fashion and written in standard English?

Reviewer #1: Yes

Reviewer #3: Yes

6. Review Comments to the Author

Reviewer #1: Thank you for reviewing your manuscript.

Ethical approval: This study involves human subjects and as such an approval number/ ethics approval must be obtained from a recognized IRB as a mandatory requirement for publication. I acknowledge that you stated you have a pending submission for approval, but this manuscript publication needs that approval.

Data: The data set available through the link you provided is not explicit enough. It doesn't show any summary statistics or appropriate data points.

Intext Citation: The style of citation used is not consistent and does not allow for ease of flow or verification and evaluation of the stated references. Please, choose a citation style and be consistent in using your preferred citation style.

Reviewer #3: Considering the importance of ethical approval before embarking on any research involving human subjects, I will still recommend a major revision of this manuscript.

Authors should provide a copy of the ethical approval.

If authors are stating that instrument's internal validity was tested in Ile-Ife, a city within Osun State Nigeria. Were data for this study collected from this location as well?

7. PLOS authors have the option to publish the peer review history of their article (what does this mean?). If published, this will include your full peer review and any attached files.

Reviewer #1: No

Reviewer #3: No

---

## [Author Response · Author response to Decision Letter 2]

26 Jun 2023

Response to Reviewers

1.This study involves human subjects and as such an approval number/ ethics approval must be obtained from a recognized IRB as a mandatory requirement for publication. I acknowledge that you stated you have a pending submission for approval, but this manuscript publication needs that approval. 

Yes. A copy of the Ethical clearance is hereby attacked to this documents.

The Ethical clearance number is IPH/OAU/12/2191. It should be note that the ethical clearance was processed after this request. Thus, it took some time to obtain the clearance. 

2.If authors are stating that instrument's internal validity was tested in Ile-Ife, a city within Osun State Nigeria. Were data for this study collected from this location as well?

Thank you for your observation, The data for the study were collected from local government area in Ife community different from area where the data for validity were sampled (I.e. Modakeke area office area).

3.The style of citation used is not consistent and does not allow for ease of flow or verification and evaluation of the stated references. Please, choose a citation style and be consistent in using your preferred citation style.

All in-text citations are now given in number reference style.

4.The data set available through the link you provided is not explicit enough. It doesn't show any summary statistics or appropriate data points.

We appreciate this observation. It should be noted that all data used for this study are now included in the datasets. The summary statistics are also added as requested. Here is the link:https://data.mendeley.com/datasets/ffffhxnjmc/1. DOI:10.17632/ffffhxnjmc.1

5.Please adjust the title either on the online submission form or in your title page so that they are identical. 

Thank you for your comment. The title of the papar has been adjusted both online and on the title page for uniformity. The title of the paper now reads “Improving food security of farming households in Nigeria: Does Broiler Outgrowers’ Programme makes any difference?”

---

## [Decision Letter · Decision Letter 3]

7 Jul 2023

PONE-D-22-26873R3Improving food security of farming households in Nigeria. Does Broiler Outgrowers’ Programme make any difference?PLOS ONE

Dear Dr. Omodara, 

Thank you for submitting your manuscript to PLOS ONE. After careful consideration, we feel that it has merit but does not fully meet PLOS ONE’s publication criteria as it currently stands. Therefore, we invite you to submit a revised version of the manuscript that addresses the points raised during the review process.

We look forward to receiving your revised manuscript.

Kind regards,

Mabel Kamweli Aworh, DVM, MPH, PhD. FCVSN

Academic Editor

PLOS ONE

Journal Requirements:

**Additional Editor Comments:**

Please kindly address all the comments raised by the reviewers and revise the manuscripts for all typographical and grammatical errors. 

Reviewers' comments:

Reviewer's Responses to Questions

**Comments to the Author**

1. If the authors have adequately addressed your comments raised in a previous round of review and you feel that this manuscript is now acceptable for publication, you may indicate that here to bypass the “Comments to the Author” section, enter your conflict of interest statement in the “Confidential to Editor” section, and submit your "Accept" recommendation.

Reviewer #1: All comments have been addressed

Reviewer #3: (No Response)

2. Is the manuscript technically sound, and do the data support the conclusions?

Reviewer #1: Yes

Reviewer #3: Partly

3. Has the statistical analysis been performed appropriately and rigorously? 

Reviewer #1: Yes

Reviewer #3: Yes

4. Have the authors made all data underlying the findings in their manuscript fully available?

Reviewer #1: Yes

Reviewer #3: Yes

5. Is the manuscript presented in an intelligible fashion and written in standard English?

Reviewer #1: Yes

Reviewer #3: Yes

6. Review Comments to the Author

Reviewer #1: The ethical statement/ clearance which was a major concern has been addressed. All major concerns have been addressed.

However, PLOS ONE does not copy edit and this manuscript needs minor editorial revisions.

Reviewer #3: Thank you for working on the last review and submitting an ethical approval for this study. However below are some clarifications and adjustments needed.

Abstract:

- Delete "well" preceding structured questionnaire. The document can either be structured, semi-structured or open-ended.

Methods:

- Indicate or qualify the type of questionnaire utilized for data collection as stated in the abstract.

- Clarify the number of out-growers and non-out-growers sampled for this study clearly as there seems to be a discrepancy between statements made in the 2nd and 3rd paragraphs of the material and methods.

- Clarify and explain the multistage sampling technique referred to in the methods section. For example, obtaining a list from CBN and Osun State Ministry of Agriculture cannot be considered a stage but purposely choosing all broiler farmers from Osun State can be a stage because a defined population was selected in this stage. Likewise, purposely selecting six LGAs from the Stage can also be considered a stage since a defined population was selected. Clearly share what population was selected at each stage showing exactly how the sampled population was selected.

Result:

- Delete "It is interesting to note" preceding the proportion of males heading each household in each study group sampled.

- Provide a credible citation for the conversion rate utilized for the Dollar to Naira rate stated in the result section.

7. PLOS authors have the option to publish the peer review history of their article (what does this mean?). If published, this will include your full peer review and any attached files.

Reviewer #1: No

Reviewer #3: No

---

## [Author Response · Author response to Decision Letter 3]

9 Jul 2023

Journal Requirements:

Thanks for your comments. All cited papers that have been retracted were done for the following reasons. 

Global Hunger Index 2021 was changed to Von Grebmer et al., 2021 who happened to be the actual authors 

Liverpool-Tasie et al., 2017 was deleted due to unavailability of reference details

There was a correction involving replacing Bijman (2016) with CBN (2016) which was the original policy document for ABOP initiative in Nigeria. The error was rectified. 

Consult E, and Ani NM (2017) was used to replace Ani (2017) due to omission of the first author.

Yamane, T. Statistics 1967 was added for the sampling size determination (in-text citation and listed reference) 

A couple of in-text citations were moved from the result section to the discussion section after being asked to separate the two sections. 

Authors such as Adepoju AO, Adejare KA (2013); Abu and Soom (2016); Emovwodo, 2019; Sanusi (2017) and Abdulazeez et al. (2018); Bature et al. (2013) and Olounlade et al. (2021) were moved. 

A thorough proofreading of the manuscript was also done to minimize possible grammatical errors. All changes were highlighted in the manuscript.

Delete "well" preceding structured questionnaire. The document can either be structured, semi-structured or open-ended.

The word “well” has been deleted

Indicate or qualify the type of questionnaire utilized for data collection as stated in the abstract. 

The questionnaire was qualified as structured 

Clarify the number of out-growers and non-out-growers sampled for this study clearly as there seems to be a discrepancy between statements made in the 2nd and 3rd paragraphs of the material and methods.

Thank you for your observation. The number of outgrowers and non-outgrowers in the 2nd paragraph of the materials and methods was well specified. On the other hand, 10 respondents each in the third paragraph addressed instrument validity only. As stated in the paper, it should be noted that these respondents were not part of the sample process for this study. In other words, they were not included in the sample size.

Clarify and explain the multistage sampling technique referred to in the methods section. For example, obtaining a list from CBN and Osun State Ministry of Agriculture cannot be considered a stage but purposely choosing all broiler farmers from Osun State can be a stage because a defined population was selected in this stage. Likewise, purposely selecting six LGAs from the Stage can also be considered a stage since a defined population was selected. Clearly share what population was selected at each stage showing exactly how the sampled population was selected.

Thank you for your comments. The multistage technique has been clearly spelt out step by step, indicating the sample techniques and populations of interest at each stage. 

Delete "It is interesting to note" preceding the proportion of males heading each household in each study group sampled.

The phrase “It is interesting to note” was deleted.

Provide a credible citation for the conversion rate utilized for the Dollar to Naira rate stated in the result section.

An official conversation rate OANDA has been quoted both in-text citation and listed in the reference

---

## [Decision Letter · Decision Letter 4]

27 Jul 2023

PONE-D-22-26873R4Improving food security of farming households in Nigeria: Does Broiler Outgrowers’ Programme make any difference?PLOS ONE

Dear Dr. Omodara,

Thank you for submitting your manuscript to PLOS ONE. After careful consideration, we feel that it has merit but does not fully meet PLOS ONE’s publication criteria as it currently stands. Therefore, we invite you to submit a revised version of the manuscript that addresses the points raised during the review process.

We look forward to receiving your revised manuscript.

Kind regards,

Mabel Kamweli Aworh, DVM, MPH, PhD. FCVSN

Academic Editor

PLOS ONE

Journal Requirements:

Reviewers' comments:

Reviewer's Responses to Questions

**Comments to the Author**

1. If the authors have adequately addressed your comments raised in a previous round of review and you feel that this manuscript is now acceptable for publication, you may indicate that here to bypass the “Comments to the Author” section, enter your conflict of interest statement in the “Confidential to Editor” section, and submit your "Accept" recommendation.

Reviewer #1: (No Response)

Reviewer #3: (No Response)

2. Is the manuscript technically sound, and do the data support the conclusions?

Reviewer #1: Partly

Reviewer #3: Yes

3. Has the statistical analysis been performed appropriately and rigorously? 

Reviewer #1: Yes

Reviewer #3: Yes

4. Have the authors made all data underlying the findings in their manuscript fully available?

Reviewer #1: Yes

Reviewer #3: Yes

5. Is the manuscript presented in an intelligible fashion and written in standard English?

Reviewer #1: Yes

Reviewer #3: Yes

6. Review Comments to the Author

Reviewer #1: I acknowledge that you used a structured questionnaire to gather primary data for this study. However, you did not state the reason for your selection of the 6 LGA's for the second stage of the multistage sampling technique.

Please clarify the statement under the subtitle 'Food security of the outgrowers and Non outgrowers households". The outgrowers food security line was N11,098.59 ($11.24) while the non outgrowers food security line was N10,188.97 ($210.31). I assume that the statement concerning the dollar value is an error.

The Reference list needs to be properly structured. Please choose a referencing style and stick to it.

I wish you all the best.

Reviewer #3: Thank you for your hard work.

Please consult with the editor to determine the acceptable English style for PLOS ONE. The entire manuscript is written in a combination of American and British English styles. Similarly, the word "Study" should not be capitalized in the first line of the Materials and Methods section. I will suggest that the manuscript be proofread once again by a native English speaker for grammatical and punctuation errors.

7. PLOS authors have the option to publish the peer review history of their article (what does this mean?). If published, this will include your full peer review and any attached files.

Reviewer #1: No

Reviewer #3: No

---

## [Author Response · Author response to Decision Letter 4]

12 Aug 2023

Reviewers’ comment/Rebuttal

1.Please clarify the statement under the subtitle 'Food security of the outgrowers and Non outgrowers' households". The outgrowers' food security line was N11,098.59 ($11.24) while the non outgrowers food security line was N10,188.97 ($210.31). I assume that the statement concerning the dollar value is an error.

Thank you for your observation. The error has been rectified. The correct figure is ($10.31)

2.I acknowledge that you used a structured questionnaire to gather primary data for this study. However, you did not state the reason for your selection of the 6 LGA's for the second stage of the multi-stage sampling technique.

I apreciate your comment. The reason for the selection of 6 LGAs was clearly stated but now modified for better understanding. It was noted that the LGs were chosen “because of the concentration of broiler growers in the area”.

3.The Reference list needs to be properly structured. Please choose a referencing style and stick to it. 

Thank you very much. We have followed the reference style of the journal for 2023 publication period. The reference list has now become properly structured.

4.Please consult with the editor to determine the acceptable English style for PLOS ONE. The entire manuscript is written in a combination of American and British English styles. Similarly, the word "Study" should not be capitalized in the first line of the Materials and Methods section. 

Thank you for your observation.The word “study” has been corrected to small letter. From further search, PLOS one accept American English. Therefore, the entire manuscript has been re-written in American English and every grammatical error and punctuation mark has been corrected. All changed or added words are highlighted in coloured ink.

---

## [Decision Letter · Decision Letter 5]

4 Sep 2023

Improving food security of farming households in Nigeria: Does Broiler Outgrowers’ Programme make any difference?

PONE-D-22-26873R5

Dear Dr. Omodara,

We’re pleased to inform you that your manuscript has been judged scientifically suitable for publication and will be formally accepted for publication once it meets all outstanding technical requirements.

Kind regards,

Mabel Kamweli Aworh, DVM, MPH, PhD. FCVSN

Academic Editor

PLOS ONE

Additional Editor Comments (optional):

Reviewers' comments:

Reviewer's Responses to Questions

**Comments to the Author**

1. If the authors have adequately addressed your comments raised in a previous round of review and you feel that this manuscript is now acceptable for publication, you may indicate that here to bypass the “Comments to the Author” section, enter your conflict of interest statement in the “Confidential to Editor” section, and submit your "Accept" recommendation.

Reviewer #1: All comments have been addressed

Reviewer #3: All comments have been addressed

2. Is the manuscript technically sound, and do the data support the conclusions?

Reviewer #1: Yes

Reviewer #3: Yes

3. Has the statistical analysis been performed appropriately and rigorously? 

Reviewer #1: Yes

Reviewer #3: Yes

4. Have the authors made all data underlying the findings in their manuscript fully available?

Reviewer #1: Yes

Reviewer #3: Yes

5. Is the manuscript presented in an intelligible fashion and written in standard English?

Reviewer #1: Yes

Reviewer #3: Yes

6. Review Comments to the Author

Reviewer #1: All concerns have been addressed. This information shared will add to the existing body of knowledge.

Reviewer #3: (No Response)

7. PLOS authors have the option to publish the peer review history of their article (what does this mean?). If published, this will include your full peer review and any attached files.

Reviewer #1: No

Reviewer #3: No

---

## [Editor Report · Acceptance letter]

11 Sep 2023

PONE-D-22-26873R5 

Improving food security of farming households in Nigeria: Does Broiler Outgrowers’ program make any difference? 

Dear Dr. Omodara:

I'm pleased to inform you that your manuscript has been deemed suitable for publication in PLOS ONE. Congratulations! Your manuscript is now with our production department. 

Kind regards, 

on behalf of

Dr. Mabel Kamweli Aworh 

Academic Editor

PLOS ONE